# Radix Glycyrrhizae Preparata Induces Cell Cycle Arrest and Induced Caspase-Dependent Apoptosis in Glioblastoma Multiforme

Tsung-Ying Lin [1], Tung-Hsuan Wu [2], Rong-Dar Tzou [1], Yi-Chiang Hsu [3,*], Kuan-Ting Lee [1]
and Tai-Hsin Tsai [4,5,6,*]

1   Division of Neurosurgery, Department of Surgery, Kaohsiung Municipal Ta-Tung Hospital, Kaohsiung 801, Taiwan
2   Division of Neurosurgery, Department of Surgery, Kaohsiung Municipal Hsiao-Kang Hospital, Kaohsiung 812, Taiwan
3   School of Medicine, I-Shou University, Kaohsiung 840, Taiwan
4   Division of Neurosurgery, Department of Surgery, Kaohsiung Medical University Hospital, Kaohsiung 807, Taiwan
5   Department of Surgery, School of Medicine, College of Medicine, Kaohsiung Medical University, Kaohsiung 807, Taiwan
6   Graduate Institutes of Medicine, College of Medicine, Kaohsiung Medical University, Kaohsiung 807, Taiwan
*   Correspondence: jenway74@isu.edu.tw (Y.-C.H.); teishin8@hotmail.com (T.-H.T.)

**Abstract:** Glioblastoma multiforme (GBM) is a highly aggressive and devastating brain tumor characterized by poor prognosis and high rates of recurrence. Despite advances in multidisciplinary treatment, GBM construes to have a poor overall survival. The Radix Glycyrrhizae Preparata (RGP) has been reported to possess anti-allergic, neuroprotective, antioxidative, and anti-inflammatory activities. However, it not clear what effect it may have on tumorigenesis in GBM. This study demonstrated that RGP reduced glioma cell viability and attenuated glioma cell locomotion in GBM8401 and U87MG cells. RGP treated cells had significant increase in the percentage of apoptotic cells and rise in the percentage of caspase-3 activity. In addition, the results of study's cell cycle analysis also showed that RGP arrested glioma cells at $G_2/M$ phase and Cell failure pass the G2 checkpoint by RGP treatment in GBM8401 Cells. Based on the above results, it seems to imply that RGP activated DNA damage checkpoint system and cell cycle regulators and induce apoptosis in established GBM cells. In conclusion, RGP can inhibit proliferation, cell locomotion, cell cycle progression and induce apoptosis in GBM cells in vitro.

**Keywords:** r Glycyrrhizae Preparata; GBM; cell cycle; cell locomotion

## 1. Introduction

Glioblastoma multiforme (GBM) is a deadly form of brain tumor in adults [1]. The median survival for GBM was only 12.6 months [2]. Current resection techniques, chemotherapeutic strategies, and radiation therapy for the treatment of GBM have developed rapidly, but these improvements have not translated into significant improvements in patient survival [3–6]. GBM is a highly aggressive and devastating brain tumor characterized by poor prognosis and high rates of recurrence [7]. Therefore, we should be more active in genetic research on the biological properties of glioblastoma, and expect to develop more potential biomarkers and therapeutic targets.

Radix Glycyrrhizae (RG) is one of the commonly used and old herbs in the East and West, and has been used since ancient times as a useful drug in traditional medicine [8,9]. In traditional herbal medicines, or to alter medicinal properties by roasting, broiling, etc., Radix Glycyrrhizae is often processed using roasting procedures, which may alter the chemical composition of licorice [10]. The roasted form of licorice (Radix Glycyrrhizae

Preparata) has been reported to possess anti-allergic, neuroprotective, antioxidative, and anti-inflammatory activities [11–14]. Some of the active ingredients in licorice have been purified, and Glycyrrhizic acid (GA) and Liquiritin(LQ) are the main active ingredients of licorice. The active ingredients in licorice has exhibit a variety of pharmacological effects, such as anti-oxidant, anti-inflammatory, anti-tumor, anti-ulcer, and anti-viral functions [15]. The anti-tumor effect of Glycyrrhizic acid has been reported in various types of tumors [16–23], through regulating several important signaling proteins, including those that belong to cysteine-dependent aspartate-specific protease (caspase) and the Bcl-2 families [24], the nuclear factor-kappa B (NF-κB) protein [25], the high mobility group box-1 (HMGB1) protein [26], the extracellular regulated protein kinases (ERK), the phosphatidylinositol 3-kinase (PI3K)/AKT kinases and the c-Jun N-terminal kinase (JNK) [24,27]. In addition, the anti-tumor effect of LQ has been reported in various types of tumor [16,27,28], through inducing cell cycle arrest and apoptosis via the ROS-mediated MAPK/AKT/NF-κB signaling pathway in hepatocellular carcinoma cells [16].LQ-triggered apoptosis was dependent on extrinsic and intrinsic pathways in cervical cancer cells through activation of Caspase-3 and poly ADP-ribose polymerase (PARP) cleavage, and Fas-associated protein with death domain (FADD)- and Bcl-2/Bax-regulated pathways, leading to Caspase-8 and Caspase-9 cleavage [27].

Notably, RGP inhibits certain cancers, including lung cancer [29]. But research on the effects of RPGs on gliomas is rather limited, and it is unclear what effects RGPs might have on GBM and how it achieves its effects. The aim of this research work was to demonstrate that RGP inhibited glioma cell viability and motion through inducing the apoptosis, to explore the possible molecular mechanisms of RGP on GBM8401 and U87MG cells by using Western Blot assay, and to provide a new insight of RGP potentially serving as an anti-glioma reagent.

## 2. Materials and Methods

### 2.1. Reagents and Chemicals

Radix Glycyrrhizae Preparata (RGP) was ordered from KO DA Pharmaceutical Co., Ltd in Taoyuan City 324, Taiwan R.O.C., Modified Eagle Medium (MEM) was purchased from Gibco, BRL (Grand Island, NY, USA). Fetal bovine serum (FBS) was purchased from Gibco, BRL (Grand Island, NY, USA). In addition, DMSO (dimethyl sulfoxide) was purchased from Sigma (St Louis, MO, USA) and MTT [3-(4,5-dimethylthiazol-2-yl)-2,5-diphenyltetrazolium bromide] was also purchased from Sigma (St Louis, MO, USA). Finally, polyvinylidene fluoride membrane (PVDF) (Millipore) was purchased from BioRad (Bio-Rad Laboratories USA). All reagents and compounds in the experiments were of analytical grade.

### 2.2. Cell Culture

Two different glioblastoma cancer cell lines, human astrocytoma U87MG (NCI-PBCF-HTB14; ATCC HTB-14) and human glioblastoma GBM8401 cells differ in the degree of malignancy. And both cell lines were purchased from Bioresource Collection and Research Center (BCRC, Hsinchu, Taiwan). GBM8401 cells were cultured in RPMI1640 medium supplemented with 10% fetal bovine serum (FBS) and 1% penicillin/streptomycin at 37 °C in a 5% $CO_2$ atmosphere. U87MG cells were cultured in modified Eagle's medium (MEM) at 37 °C in a 5% $CO_2$ atmosphere supplemented with 10% FBS and 1% penicillin/streptomycin.

### 2.3. Cell Viability in GBM 8401 Cell and U87MG Cell after RGP Treatment

Cell viability was tested using the MTT assay described by Mossman (1983) [30]. The GBM8401 cell and U87MG cell ($3 \times 10^4$) were seeded to 24-well plate (0.1 mL of medium per each well) in cultured overnight and cells were suspended in culture medium containing 10% FBS; and incubated in an atmosphere containing 5% $CO_2$, saturated humidity, and 37 °C for 24 h. The next day, the cells were treated with 250–200 mg/mL RGP and incubated with 3-(4,5-dimethylthiazol-2-yl)-2,5-diphenyltetrazolium bromide (MTT) assay for 3 h.

After 3 h of incubation, DMSO was added to stop the reaction and optical density was determined at 540 nm with a multi-well plate reader (Powerwave XS, Biotek LabX, Midland, ON, Canada). The value of IC50 was determined. When there were no cells, we subtracted the background absorbance of the medium.

### 2.4. The Synergistic Therapeutic Effect of RGP and Radiation

In order to confirm the synergistic therapeutic effect of RGP and radiation, different cell numbers were implanted according to different radiation intensities, 0 Gy: 100 cells, 1 Gy: 200 cells, 2 Gy: 400 cells, 4 Gy: 1000 cells. After the cells were seeded, treated with RGP 1 mg/mL to irradiate at different intensities and placed in a cell incubator for 7 days. The medium was changed every two days during this period. After 7 days of culture, cells were fixed with 4% formaldehyde for 15 min. Then, after staining with 0.5% crystal violet, the cells were washed with water and then with crystal violet, and finally the number of formed cell colonies were counted. These numbers are used in the formula to calculate the survival fraction (SF). Plating efficiency was defined as the percentage of cells that grew into colonies. PE = counted colonies/inoculated cells. SF = counted colonies/inoculated cells.

### 2.5. Cell Cycle Analysis

Cell cycle analysis to analyze the effect of drug treatment on the cell cycle and describe by previous study [31]. We placed U87MG and GBM8401 cells in a cell incubator, and we placed the cell density ($1 \times 10^6$) in 6-well plates and cultured overnight. After 24 h of incubation, the cells were centrifuged in centrifuge tubes and the supernatant was collected. After removing the supernatant, wash twice with PBS and rinse with $1\times$ trypsin. After rinsing, place them in a 37 °C oven for 1–2 min to detach the cells. After standing in the oven for 1 min, centrifuge again to remove the supernatant, set the centrifuge at $2500\times g$ rpm and run for 5 min, collect them in a 5 mL centrifuge tube, centrifuge for 5 min, remove the supernatant, and leave the cell culture liquid. Then perform a third centrifugation, add 1 mL of PBS to the remaining culture medium to wash, set the centrifuge to $2500\times g$ rpm for 5 min, add 500 µL PBS to disintegrate the cell tray, after centrifugation, in order to fix the cells, slowly 500 µL of 70% ethanol was added to the medium and placed in the refrigerator for fixation. After overnight fixation, a fourth centrifugation was performed. The centrifuge was set to $2500\times g$ rpm for 5 min. After centrifugation, the ethanol-containing supernatant was removed and the cells were washed with 1 mL of PBS. And 5 µL of RNAse A 100 mg/mL was added, placed in a 37 °C oven, and reacted for 30 min. For the detection of apoptosis assay, nuclei of DNA were stained with propidium iodide, so 20 µL of propidium iodide 2 mg/mL (final concentration 40 µg/mL) was added and the cells were placed in an oven at 37 °C 15 min. Finally, cells were transferred from centrifuge tubes to Falcon tip centrifuge tubes and samples were mounted on an analytical machine (Beckman coulter FC500 and FACSCalibur, BD, USA). WinMDI 2.8 free software (Mississauga, BD, USA) is required. The analysis software can be opened, plotted, and analyzed statistics.

### 2.6. Assessment of Apoptosis

The apoptotic rates were analyzed as described [32,33]. with slight modification, by using the FACS methodology using a protocol recommended in the Annexin V-FITC Apoptosis Detection Kit I according to the manufacturer's instructions. Cells were first cultured in 6-well culture plates (Orange Scientific, EU). And RGP was used to treat GBM8401 cell and U87MG cell. After four hours of treatment, the drug-treated cells were harvested by centrifugation. These received cells were added at room temperature containing Annexin V-FITC and propidium iodide. (PI) (100 mg/mL) in $1\times$ Annexin binding buffer, protected from light for 15 min, Finally, cells were transferred from centrifuge tubes to Falcon tip centrifuge tubes and samples were mounted on an analytical machine (Beckman coulter FC500 and FACSCalibur, BD, USA). WinMDI 2.8 free software (Mississauga, BD, USA) is required. The analysis software can be opened, plotted, and analyzed statistics.

*2.7. Caspase3 Activity Assay*

The activity of caspase-3 was detected with Caspase-3 (activity) FITC staining kit according to the manufacturer's instructions [34,35]. GBM8401 cells and U87MG cells were seeded into 6-well plates ($1 \times 10^6$/mL) and incubated overnight. The cells were treated with different concentrations of RGP (0, 0.25, 0.5, 1 mg/mL) for 24 h. After 24 h, the culture medium containing the cells was collected in a 15 mL centrifuge tube and washed with PBS. Next, add Caspase-3 antibody to each tube and incubate for 30 min. The first centrifugation, after centrifugation at $2500 \times g$ rpm for 5 min, after removal of the supernatant, a second wash with PBS, the cells were resuspended in 0.5 mL PBS and centrifuged again. Finally, cells were transferred from centrifuge tubes to Falcon tip centrifuge tubes and samples were mounted on an analytical machine (Beckman coulter FC500 and FACSCalibur, BD, USA). WinMDI 2.8 free software (Mississauga, BD, USA) is required. The analysis software can be opened, plotted, and analyzed statistics.

*2.8. Migration Assay*

Migration assay is used to measure cell motility. We used Transswell chambers (BD Falcon™, San Jose, CA, USA) to measure cell motility. GBM8401 and U87-MG cells were inserted into 24-well cell culture plates, and the number of cells implanted such as U87MG ($1 \times 10^6$) and GBM8401 ($5 \times 10^5$). After 24 h of culture, cells were treated with RGP (0, 0.25, 0.5, 1 mg/mL) and cultured for an additional 24 h. The next day, cell numbers were observed under the microscope at 0, 12, 24, and 48 h.

*2.9. Invasion Assay*

Invasion assays measure cell invasive capacity and perform in vitro assays in Transswell chambers (COR3452; CORNING, Corning, NY, USA). GBM8401 and U87-MG cells were seeded at $1 \times 105$ cells in 300 μL of serum-free medium and 1 mL of medium was added to the lower chamber of each Transwell. After 24 h of culture, the cells were treated with the therapeutic drug RGP (0, 0.25, 0.5, 1 mg/mL) for 24 h. After an additional 24 h incubation, the remaining cells on the surface of the Transswell membrane were removed with a cotton swab. The insert chambers were fixed and stained with 0.5% crystal violet, washed with water after staining, and images were taken under the microscope. Count the number of cells in six random high-power fields.

*2.10. Adhesion Assay*

Adhesion assays are used to test the ability of specific glioma cells to adhere to adhesive substrates. The GBM8401 and U87MG cells were washed with PBS, $1\times$ Trypsin was added, and they were placed in 37 °C oven for 1–2 min. When the cells fell off, we used new culture medium to collect supernatant into a 15 mL centrifuge tube. Cells were seeded to 24-well plates (0.1 mL of medium per each well) at $3 \times 10^4$ cells/well density. The next day, cells were fixed with 4% formaldehyde for 15 min and washed with PBS, and then stained with 0.5% crystal violet (Sigma-Aldrich; Louis, MO, USA), rinsed with water, washed the crystal violet, and photographed under a microscope.

*2.11. Western Blotting*

Western blotting was performed as described [36]. Logarithmic growth phase of $1 \times 10^6$ cells i.e., GBM8401 and U87MG were inoculated on Petri dishes (100-mm). Next, either vehicle or GRP treated the cells. Then the cells washing were carried out with cold PBS, followed by solubilizing in 200 μL of lysis buffer with phosphatase and protease inhibitors. The protein separation using a total of 80–100 μg cell lysates were used by 10–12% SDS-PAGE and then transferred to PVDF membranes and subjected to electrophoresis at 50 V for 4 h. PVDF membranes were blocked with 5% non-fat milk under the condition of overnight. Then, the PVDF membranes were incubated with primary antibodies [Cyclin A2 (1:1000; proteintech; 18202-1-AP), Cyclin B1 (1:1000; proteintech; 55004-1-AP), CDK1 (1:1000; cell signaling; E1Z6R), CHK2 (1:1000; abgent.com; AP4999a), p-CHk2 (1:1000; ab-

gent; AP50241), CHK1 (1:1000; proteintech; 22018-1-AP), p21 (1:1000; Cell Signaling; #2947), and β-actin (1:20,000; Sigma; A5441)] for 2 h at room temperature. Subsequently, the membranes were washed several times and incubated with a corresponding secondary antibody (IRDye Li-COR, Lincoln, NE, USA) at a dilution of 1:20,000 for 30–45 min. Antigens were then visualized using a near-infrared fluorescence imaging system (Odyssey LICOR, USA), and the data were interpreted using the Odyssey 2.1 software LI-COR Biosciences - U.S. or a chemiluminescence (ECL; Amersham Corp., Arlington Heights, IL, USA) detection reagents. Densitometry analysis (including integrated density of bands) was carried out via Image J (NIH), followed by normalizing the documented values to beta-actin.

### 2.12. Mitotic Index Analysis

To differentiate between G2 arrest and mitotic arrest, we used a mitotic maker MPM-2 (antiphospho-Ser/Thr-Pro) to analyze the mitotic index of cells after drug treatment [37]. The antibody can be used as an indicator of mitotic dysfunction because it recognizes proteins that are fully phosphorylated during mitosis. And using nocodazole (15 μg/mL), a metaphase arrest inducer, as a positive control, we treated different groups of GBM8401 and U87MG cells with nocodazole (15 μg/mL). MPM-2 was performed by harvesting and fixing cells for 24 h, washing the cells with MPM-2 antibody at room temperature every other day, and resuspending in 100 μL of IFA-Tx buffer (4% FCS, 150 nM NaCl, 10 nM HEPES, 0.1% sodium azide, and 0.1% Triton X-100) for 1 h. Cells were then washed again and resuspended in IFA-Tx buffer with rabbit anti-mouse FITC-conjugated secondary antibody (1:50 dilution; Serotec) for 1 h at room temperature in the dark. Finally, cells were washed and resuspended in 500 μL of PBS with 20 μg/mL PI (Sigma) for 30 min in the dark. The degree of MPM-2 expression was measured by FACSCalibur flow cytometry. Then WinMDI 2.9 was used to analyze the performance of MPM-2 antibody in the experimental group and the control group.

### 2.13. Data Analysis

Data are expressed as the mean ± standard error of the mean of at least three independent experiments. One-way and two-way analysis of variance with GraphPad prism's test were used for statistical analysis. A *p* value of <0.05 was considered statistically significant. All statistical operations were performed using SPSS 24.0 software (SPSS, Inc., Chicago, IL, USA).

## 3. Result

### 3.1. RGP Suppressed the Cell Viability of GBM8401 and U87MG Cells

To evaluate the cytotoxic effect of RGP on human brain malignant glioma cells (GBM8401 and U87MG), cells were treated with various concentrations of RGP (0, 0.25, 5, 1 mg/mL) for 24 h followed by cell viability measurements using the MTT assay, respectively. As can be seen in (Figure 1), while there was no change in normal lung fibroblasts MRC-5 cells (data not shown). The cellular proliferation of GBM8401 and U87MG cells was significantly decreased following RGP treatment in a dose-dependent manner (U87MG; $y = -11.494x + 110.56$, $R^2 = 0.9844$, GBM8401; $y = -18.343x + 121.23$, $R^2 = 0.9756$), indicating that RGP dose-dependently inhibited clone formation of GBM8401 and U87MG cells. The inhibitory effects of RGP were similar between the GBM8401 and U87MG cells. The growth inhibitory dose of 50% (ID50) of RGP was 3.88 mg/mL in GBM8401 cells and 5.26 mg/mL in U87MG cells, respectively.

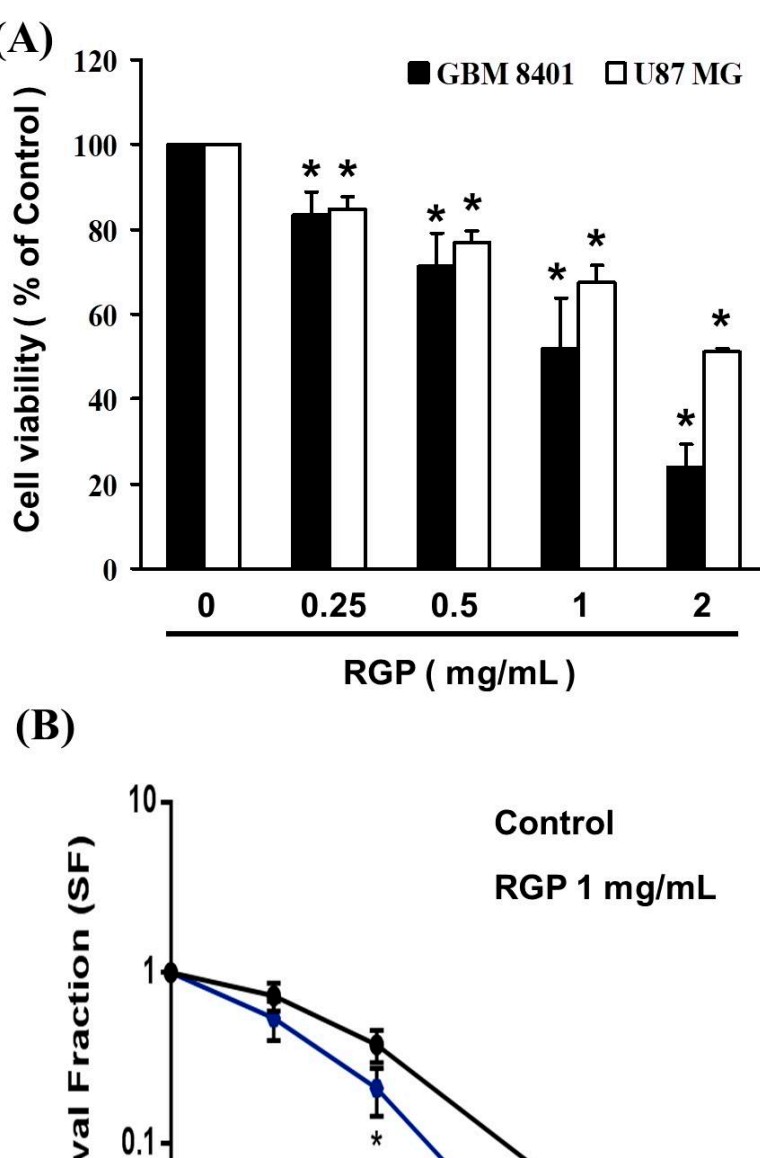

**Figure 1.** RGP mediates the cell viability of GBM8401 and U87MG cells. (**A**) Cells were treated with increasing doses of RGP for 24 h in vitro. (**B**) The GBM8401 cells were co-treated with increasing doses of RGP combined with radiation in vitro. * $p < 0.05$.

In addition to this, we also wanted to evaluate the role of synergistic therapy with RPG and Radiation on GBM8401 and U87MG cells. As shown in (Figure 1B), the results of our cell viability analysis by MTT assay, growth was significantly suppressed by co-treatment with increasing doses of RGP and radiation.

### 3.2. RGP Induced the Caspase-Dependent Apoptosis of GBM8401 Cells

As Figure 1 show the decrease in the cell viability of GBM8401 and U87MG cells could be related to cell death. Therefore, we analyzed cell death using the Annexin V-FITC detection and propidium iodide (PI) staining to confirm whether RGP caused cytotoxic effects on glioblastoma cells. This evaluation showed an increase in GBM8401 cell death after treatment with RGP (0, 0.5, 1, 2 mg/mL) for 4 h (Figure 2). Cell populations and apoptotic ratios were analyzed through flow cytometry. To analyze cell populations we use flow cytometry. And through PI staining to distinguish whether the cells are necrotic or apoptotic, the results of Annexin-FITC/PI assay showed that apoptotic cells under RGP treatment and without RGP treatment showed significant changes, that is, compared with untreated cells, the percentage of apoptotic cells was significantly increased in the treated cells (Figure 2B; $y = 0.393x + 1.585$ $R^2 = 0.6094$), and results show that RGP can induce glioblastoma apoptosis but not necrosis.

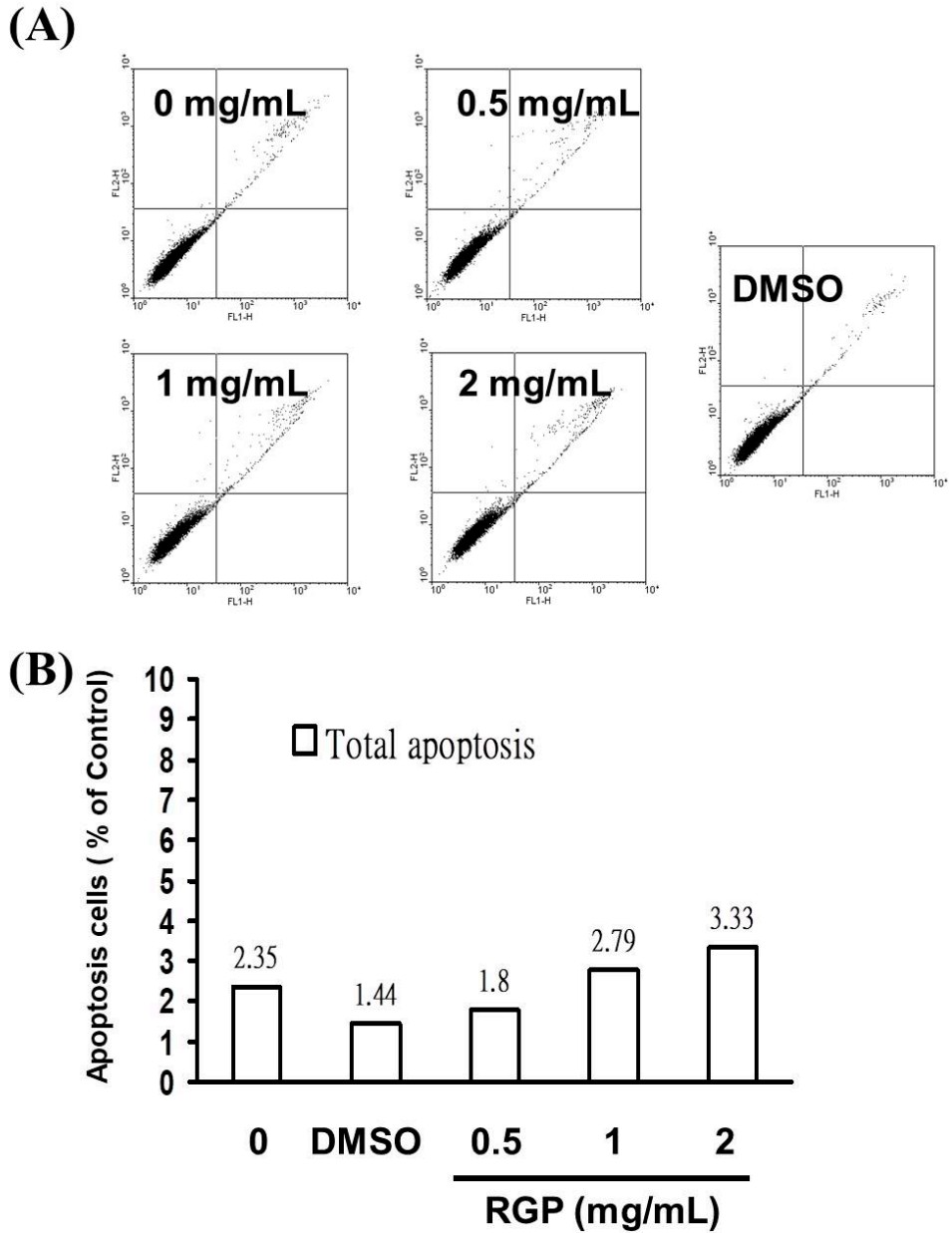

**Figure 2.** RGP induced apoptosis in GBM8401 cells. (**A**) Annexin V-FITC and (**B**) PI staining.

Subsequently, caspase-3 assay was performed to understand whether cell death was being triggered by caspase-dependent apoptosis. As the concentration of RGP increased (0–1 mg/mL), the number of active caspase-3 cells increased linearly. A significant rise in the percentage of caspase-3 activity was detected in RGP-treated cancer cells (Figure 3) [$y = 2.761x - 1.69$ $R^2 = 0.8512$]. We suggested that RGP induced GBM8401 cell death by caspase-3-dependent apoptosis.

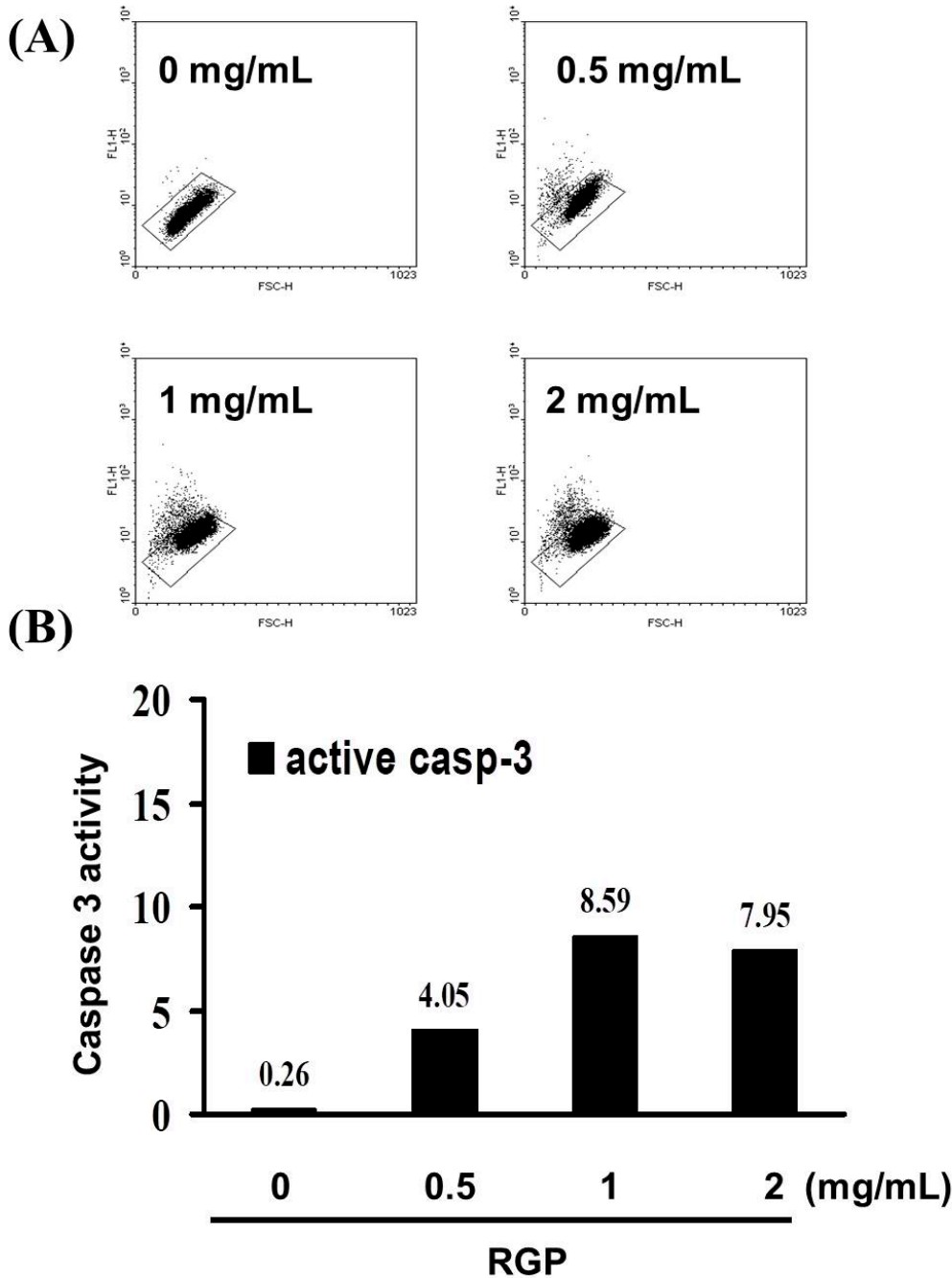

**Figure 3.** (**A**) Increases in active caspase-3 cells induced by RGP treatment in the GBM8401 cell line. (**B**) Results are expressed as a percentage of the controls, with controls being 100%.

### 3.3. RGP Attenuates the Migration, Invasion and Adhesion of Glioma Cells

To talk about the role of RGP in GBM cell motility, therefore Matrigel invasion assay, migration assay and adhesion assay were used to analyze the invasion, migration and adhesion of the GBM8401 cells following treatment with RGP. After incubation with differ-

ent concentrations of RGP (0, 1 mg/mL), the in adhesion, migration and adhesion of the GBM8401 cells were obviously inhibited. We also found that the migration was significantly inhibited in (GBM8401 and U87MG) cells after being treated with RGP (Figure 4A,B). Matrigel invasion assay showed cell invasion to be suppressed in both U87MG and GBM8401 cells (Figure 5A,B). In addition, our cell adhesion assay found marked inhibition of the cell lines treated with RGP (1 mg/mL) at 24 h (Figure 6A,B). These results indicate that RGP reduced the cell invasion, migration and adhesion of glioma cells in vitro.

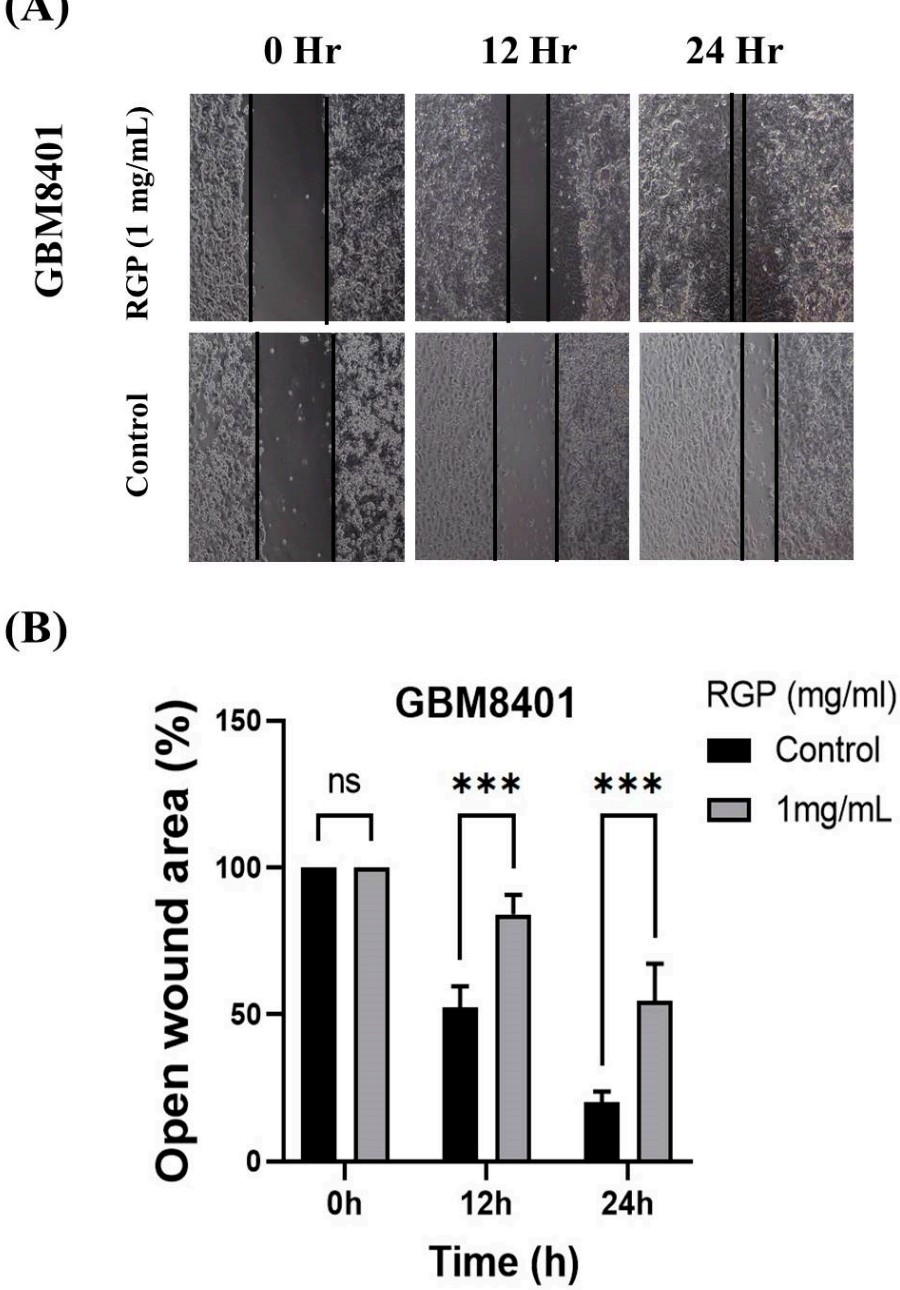

**Figure 4.** RGP inhibits the migration ability of GBM cell line. (**A**) GBM cell lines using different concentrations RGP (0, 1 mg/mL) for 24 h. (**B**) RGP reduced the migration ability of GBM cell lines. Statistical analysis used the *t*-test, with significance set at *** $p < 0.001$ compared with the control group.

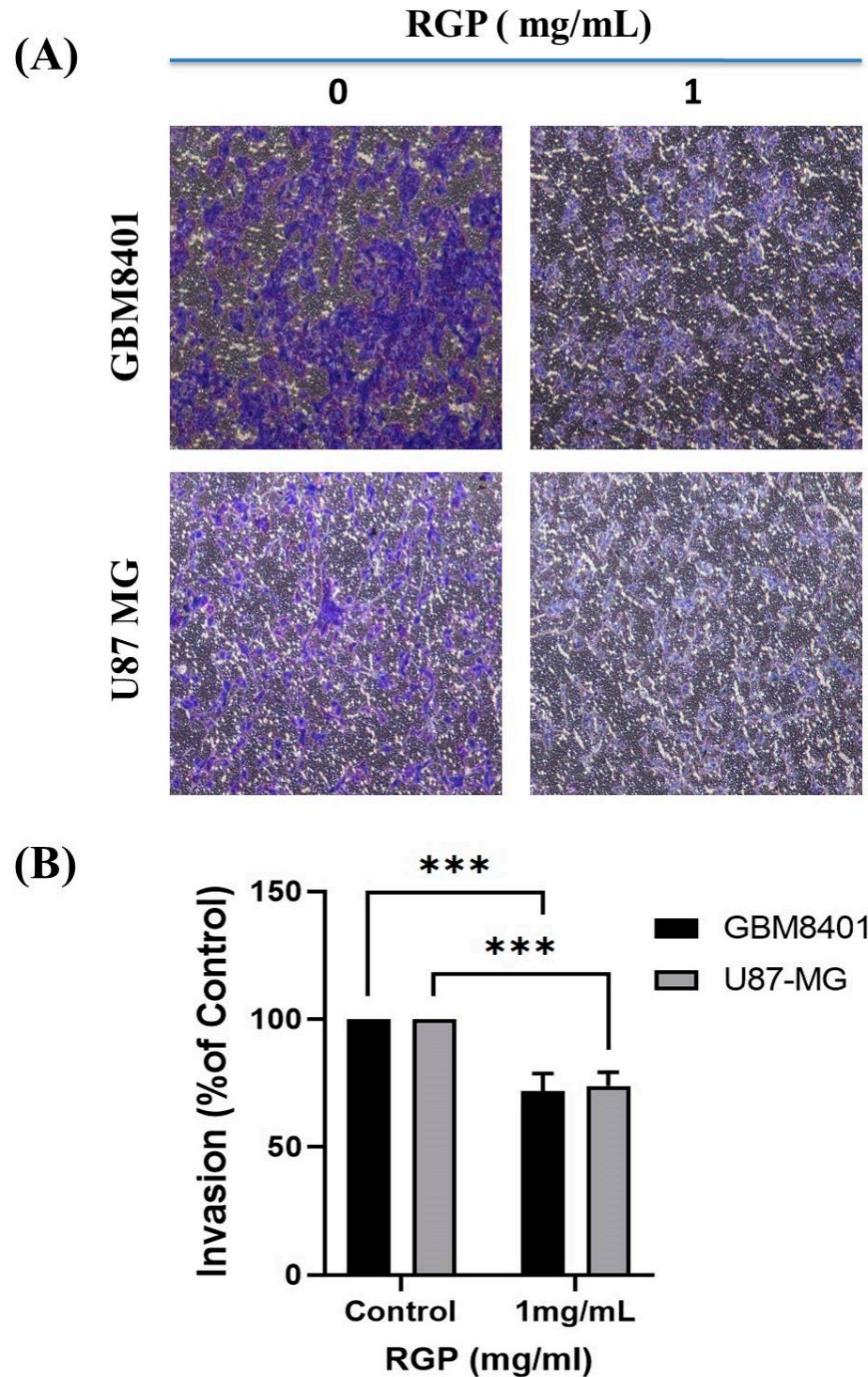

**Figure 5.** RGP inhibits the invasion ability of GBM cell line. (**A**) GBM8401 and U87 MG cell lines were using different concentrations RGP (0, 1 mg/mL) for 24 h. (**B**) RGP (1 mg/mL) reduced the invasion ability of GBM8401 and U87 MG cell lines. Statistical analysis used the *t*-test, with significance set at *** $p < 0.001$ compared to controls.

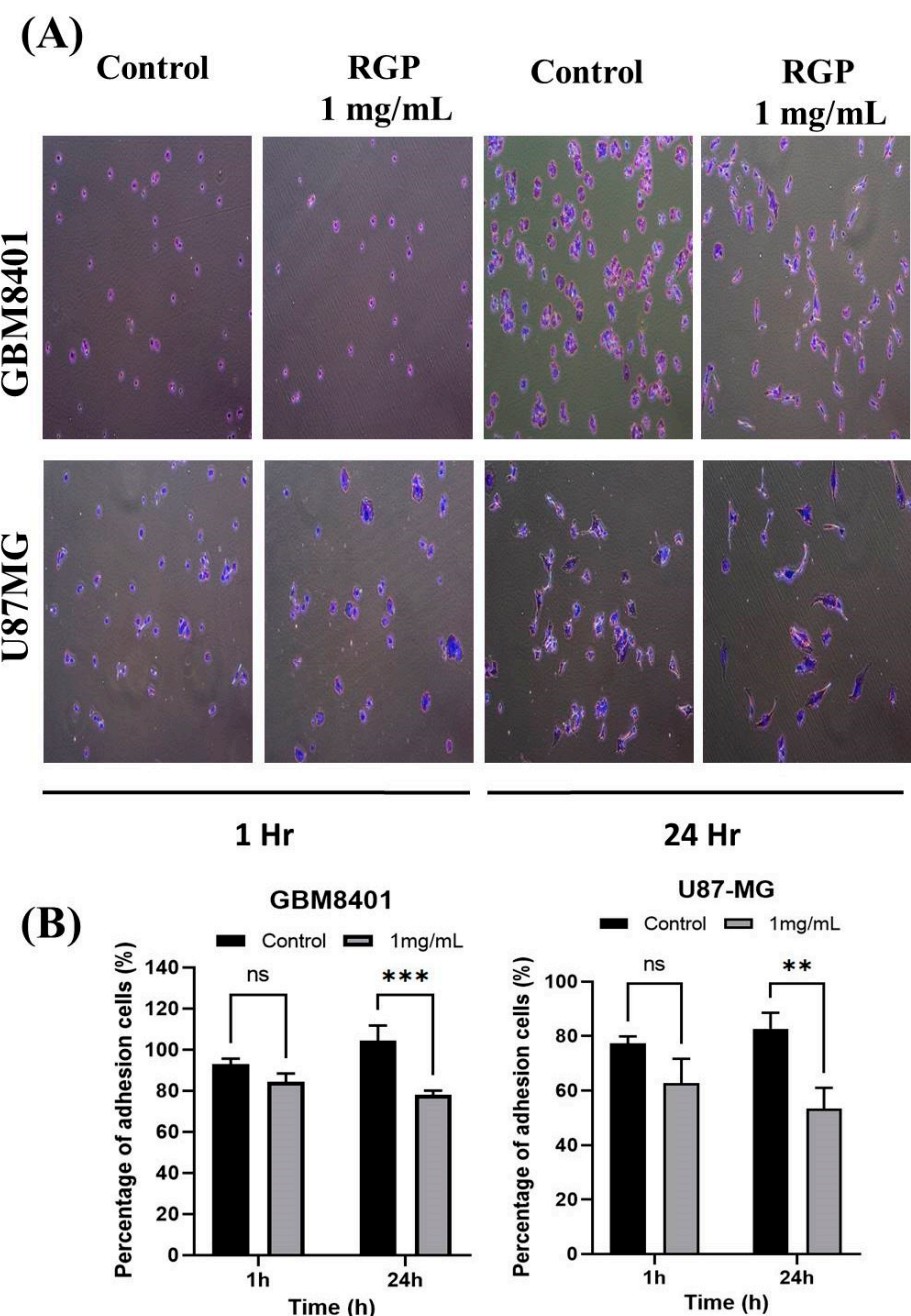

**Figure 6.** Influence of RGP on adhesion ability of GBM cell lines. RGP reduced the adhesion ability of (**A**) U87MG and GBM8401 cells. (**B**) Adhesion assay found RGP (1 mg/mL) to have markedly inhibited at adhesion at 24 h. Student *t*-test was used to test data, with significance set at ** $p < 0.01$, *** $p < 0.001$ compared to controls.

### 3.4. RGP Triggers $G_2/M$ Cell Cycle Arrest in U87MG and GBM8401 Cells

To determine the impact of RGP treatment on cell cycle phase, GBM8401 and U87MG cells were treated with RGP and the cycle progression was examined by cell cycle analysis. When cells were exposed to different concentrations of RGP (1 mg/mL; Figure 7A) and for different lengths of time (0 h, 8 h, 16 h, 24 h, 48 h, 72 h; Figure 7B), we observed that RGP increased the number of cell populations in the $G_2/M$ phase (Figure 7A). Our results indicated that exposure to RGP resulted in a dose-dependent increase in the number of cells in the $G_2/M$ phase, suggesting a reduction in mitosis in U87MG and GBM8401, though cell cycle did not change over time. The results imply that RGP induced GBM8401 and U87MG cells apoptosis could though the G2/M cell cycle arrest.

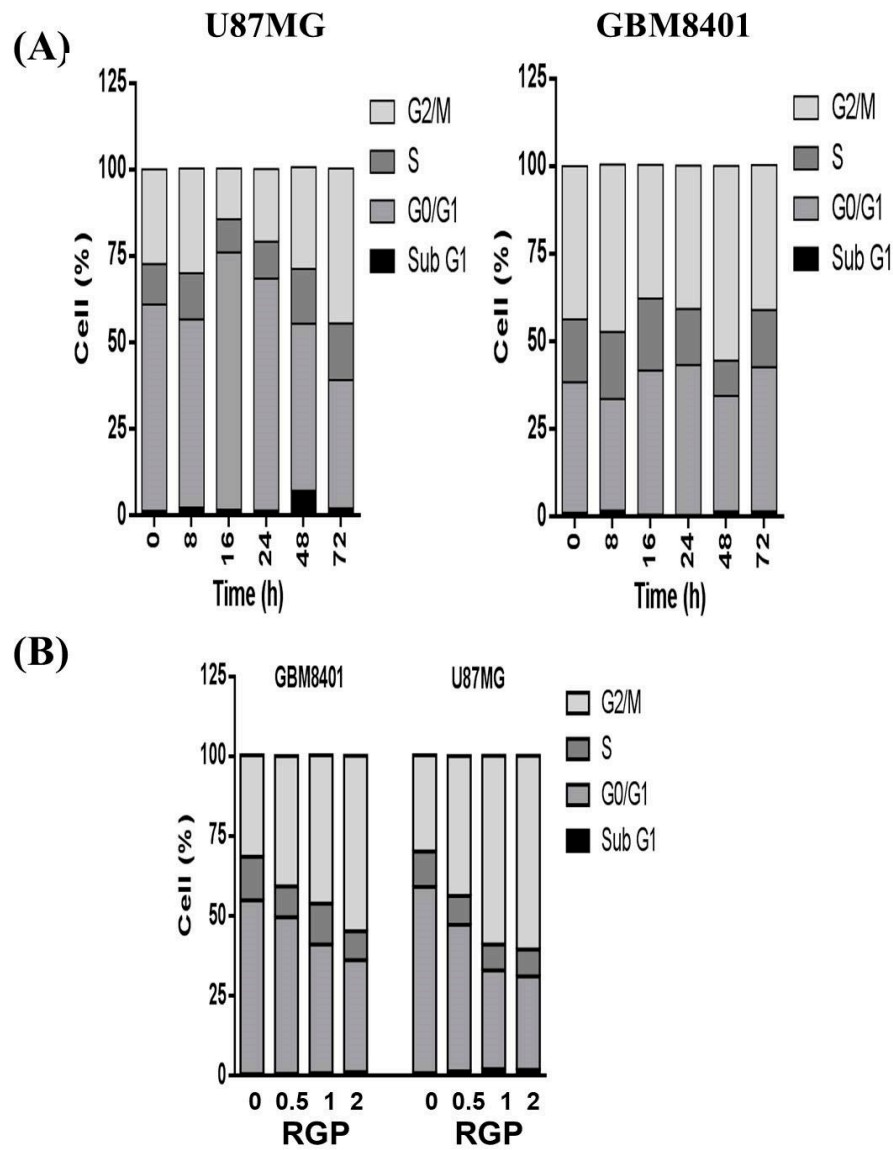

**Figure 7.** Effects of RGP on cell cycle progression/distribution in GBM cell lines. (**A**) Cells were exposed to RGP (1 mg/mL) for different lengths of time (0 h, 8 h, 16 h, 24 h, 48 h, 72 h) (**B**) Cells were exposed to different concentrations of RGP (0, 0.5, 1, 2 mg/mL). RGP induced an increase in S and G2/M-phase cell percentages (%). Cells were stained with propidium iodide to analyze DNA content, quantified by flow cytometry.

To distinguish G2 arrest from mitotic arrest, MPM-2 stain was used to analyze the mitotic index after RGP treatment. As we known, MPM-2 is commonly used as an indicator of mitotic disturbance. Figure 8 displays the number of cells exposed to RGP at various concentrations RGP (0, 0.5, 1, 2 mg/mL). Figure 8A,B demonstrate that exposure to RGP increased levels of protein synthesis during mitosis (y = 10.465x + 13.965 $R^2$ = 0.4341). This antibody is able to recognize proteins whose epitopes are during mitosis (Glatz et al., 2010). Nocodazole (15 µg/mL) is an inducer of metaphase. We treated different groups of GBM 8401 cells with nocodazole (15 µg/mL) to provide a positive control (Seo et al., 2006). These results suggest that regarding increased protein synthesis during mitosis in the MPM-2 staining. It is implied that cell pass the G2 checkpoint without cell cycle arrest in RGP Cells.

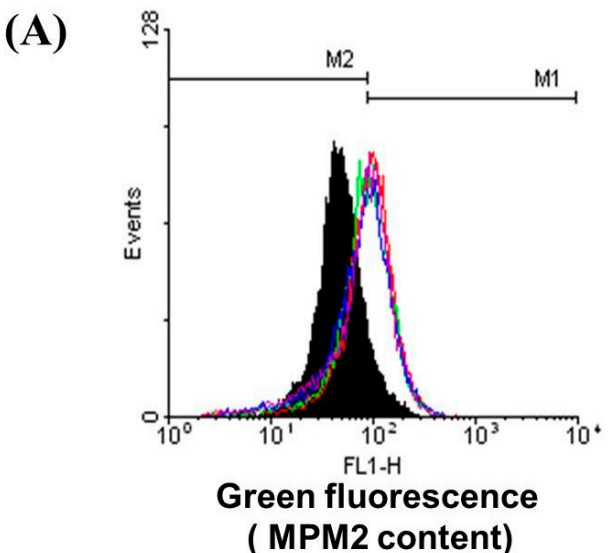

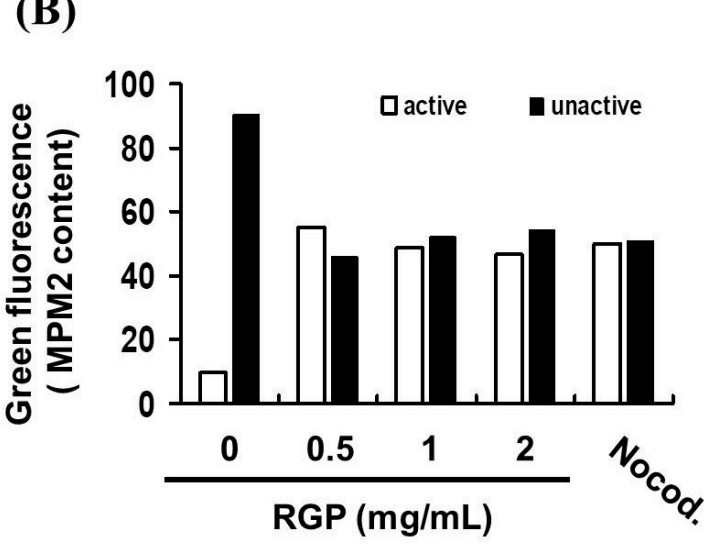

**Figure 8.** Mitotic index was assessed based on MPM-2 expression. Flow cytometry analysis of MPM-2 expression in the treated cells was conducted. (**A**) Cells were either treated or not treated with 0, 0.5, 1, 2 mg/mL RGP. After 24 h of treatment, cells were fixed with 70% ethanol, stained with MPM-2 and PI, and analyzed using FACScan software. (**B**) Results are expressed as percentage of the controls, with the controls being 100%.

### 3.5. RGP Modulated the Expression of Cell Cycle-Related Proteins

To investigate the mechanism of RGP against GBM8401 and U87MG cells, the expression levels of cell cycle-associated proteins were individually detected by using Western blot analysis (Figure 9). We measured the relative intensities of cell cycle regulators such as CylcinA2, CyclinB1, CDK1; positive checkpoint kinases such as p-CHK2, CHK1/2; and negative checkpoint kinase such as p21. As (Figure 9) shown, it was found that the expression level of the negative checkpoint kinases such as p21 was increased. In addition, RGP down-regulated protein expressions of CylcinA2, CyclinB1, CDK1. Moreover, RGP upregulated the expression of the checkpoint kinases such as p-CHK2, CHK2.

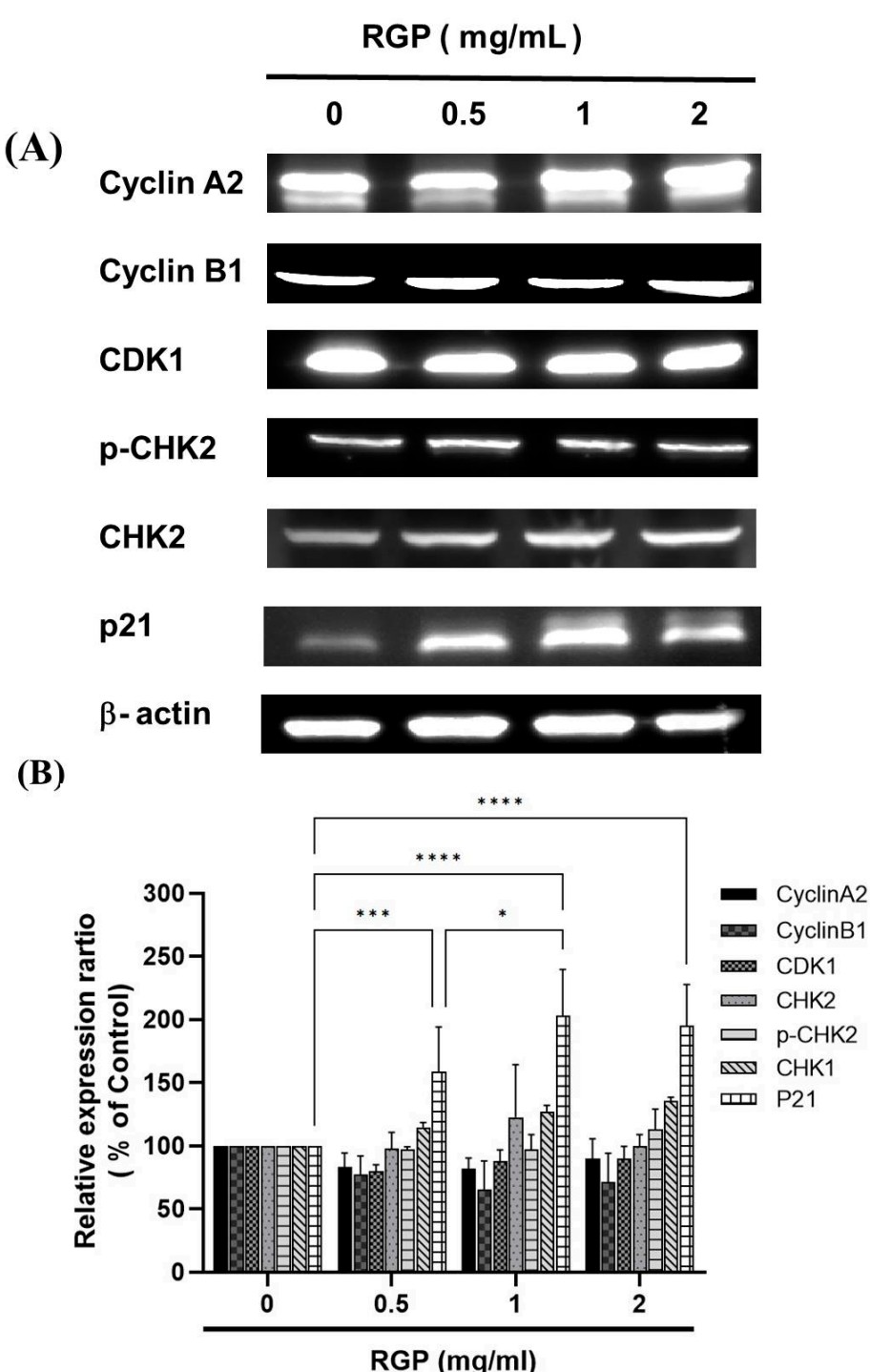

**Figure 9.** (**A**) RGP regulated DNA damage checkpoint kinase expression in CyclinA2, CyclinB1, CDK1, p-CHK2, CHK1/2 and p21gene expression in the GBM8401 cells. Cells were treated with RGP (0, 0.5, 1, 2 mg/mL) for 24 h. Gene and protein expression was subsequently detected by Western blotting. (**B**) RGP significantly increased p21 expression. Results are expressed as a percentage of the controls, with controls being 100%. Student *t*-test was used to test data, with significance set at * $p < 0.01$, *** $p < 0.001$, **** $p < 0.0001$ compared to controls.

## 4. Discussion

This study clearly states that RGP attenuates viability and reduces motility in GBM8401 and U87MG cells. A significantly increased the percentage of apoptotic cells and the percentage of caspase-3 activity. In addition, the cell cycle analysis demonstrates that cells

expose to RGP could not pass the G2 checkpoint, resulting in G2/M phase arrest. Based on the results, it appears to imply that RGP activates the DNA damage checkpoint system and cell cycle regulators leading to cell cycle arrest; and RGP induces caspase-dependent apoptosis in established GBM cells.

Most of the chemo-preventive agents currently used in clinical oncology utilize an intact apoptosis signaling pathway to trigger cell apoptosis. Apoptosis mainly consist of two main pathways and third is executioner pathway of apoptosis. The intrinsic pathway is mediated by Bax/Bak insertion into mitochondrial membrane, and subsequently, cytochrome c released which combines with Apaf-1 and procaspase-9 to produce apoptosome followed by the activation of caspase-3 cascade of apoptosis [38]. The extrinsic pathway triggered by external stimuli or ligand molecule and particularly involves death receptors [39]. Both extrinsic and intrinsic pathways converge at execution phase [40], then activates executioner caspases such as caspase-3, caspase-6 and caspase-7, Caspase-10 [41]. The understanding of the apoptotic pathway and how tumor cells evolve apoptosis to against cell death has focused research on new strategies aimed at inducing cancer cell apoptosis [42]. It is known that RGP can induce apoptosis in cancer cells. In 2004, Jo et al. proposed that licorice root may have chemopreventive effects against human breast cancer through the modulation of the expression of the Bcl-2/Bax family of apoptotic regulatory factors [43]. It also showed that The G. uralensis exhibited estrogenic effects in breast cancer cells, results were associated with up-regulation of tumor suppressor gene p53 and pro-apoptotic protein Bax [44]. This result is similar to the conclusion of our experiment. Based on the results of this experiment, RGP has no cytotoxic effect on Hs-68 cells (normal skin fibroblasts) but has a cytotoxic effect on GBM8401 cells on the contrary. As Figure 2 show, the Annexin-FITC/PI assay revealed that the apoptotic cells increased significantly. Moreover, a significant rise in the percentage of caspase-3 activity was detected in RGP-treated cancer cells (Figure 3). Therefore, according to the effects of RGP in the upregulation of caspase-3 and inducing the apoptosis, we suggested that RGP causes apoptosis by activating the caspase pathway.

The uncontrolled cell cycle of cancer cells is a feature of cancer. It involves mutations in genes related to cell cycle regulation [45]. Mutations in these related genes enable cancer cells to skip the supervision of the checkpoint control system and strengthen checkpoint control. Systematic surveillance is one of the potential strategies for current anticancer drug therapy. To the best of current knowledge, treatment with RGP can induce cell cycle arrest in some types of cell lines. Previous literatures have shown that RGP induces tumor cell cycle arrest. Jo et al. demonstrated that G. uralensis extract caused the up-regulation of p21 and down-regulation of CDK2 and cyclin E and induced G1 cell cycle arrest [44]. In addition, Seon et al. determine whether and by what mechanism the hexane/ethanol extract of G. uralensis and its active component, isoangustone A, inhibit cell-cycle progression in human prostate and mouse breast cancer cells [46]. Isoangustone A dose-dependently decreased DNA synthesis and induced G1 phase arrest in human prostate and mouse breast cancer cells, reduced the levels of CDK2 and CDK4 as well as cyclin A and cyclin D1 proteins, and also induced a decrease in CDK2 activity. These results demonstrate the potential of hexane/ethanol extract of G. uralensis containing isoangustone A as an antitumor agent. Also Rafi et al. demonstrate that licorice root contains beta-hydroxy-DHP, which induced Bcl-2 phosphorylation, apoptosis, and G2/M cell cycle arrest, in breast and prostate tumor cells [47]. Based on the results, RGP induced an increase in the number of cells in the G2/M phase (as shown in Figure 6). When treating with RGP treatment, the cells could be supervised by the G2 checkpoint (as shown in Figure 7) and then induced cell cycle arrest in GBM8401 cells. Overall, exposure to RGP that failed to pass the G2 checkpoint resulted in cell cycle arrest in the G2/M phase in GBM8401 cells, ultimately leading to subsequent apoptosis in cancer cells.

Dysregulation of the cell cycle is a feature of human cancers and underlies abnormal cell proliferation, and loss of cell cycle checkpoint control promotes genetic instability [48]. The cell cycle is controlled by many cell cycle control factors [49], namely positive regulatory molecules, negative regulator molecules, which are regulators of the cell cycle, activate or

inhibits cell cycle factors that are essential for the start or stop of the next cell cycle phase [50]. Therefore, these regulatory molecules either promote progress of the cell to the next cycle or halt the cycle. Positive regulator molecules such as cyclins and cyclic-dependent kinases, are responsible for the progress of the cell through the various checkpoints [51]. Negative regulator molecules such as retinoblastoma protein (Rb), p53, p21, CHK1 and CHK2, monitor cellular conditions and can halt the cycle until specific requirements are met [52]. Activated cell cycle checkpoints oversee functions, thereby delaying cell cycle progression and promoting DNA repair. Furthermore, DNA checkpoint surveillance systems can also protect the organism from cancer by inducing cell death to remove harmful damaged cells [53]. There are currently two pathways that primarily involve checkpoint Christ, such as the ATM-CHK2-p53 pathway controlling the G1 checkpoint or the ATR-CHK1-Wee1 pathway controlling the S and G2/M checkpoints [20,54,55]. When these DNA checkpoint systems are activated, it regulates positive and negative regulatory molecules including cyclins, CDKs, CHK1/CHK2 and p21 and then delays cell progression or death. To understand why cell cycle arrest occurs after RGP treatment, we measure the relative intensities of cell cycle regulators such as cyclins, cyclic-dependent kinases, and checkpoint kinases. The relative intensities of CylcinA2, CyclinB1, and CDK1 were significantly down-regulated upon GRP treatment; the relative intensities of CHK2 and p-CHK2 were significantly up-regulated in RGP-treated GBM8401 cells. Furthermore, the strength of p21 as a cyclic-dependent kinase 1 inhibitor was also significantly upregulated in RGP-treated GBM8401 cells. Based on the above results, it is clearly suggested that RGP activates positive checkpoint kinases such as CHK1, CHK2 and p-CHK2, thereby inhibiting the expression of CDKs and cyclins; RGP also activates negative checkpoint kinases such as p21, thereby inhibiting the expression of CDKs and cyclins, resulting in the inability of cells to carry out cellular processes, eventually leading to cell cycle arrest or inducing apoptosis.

In Chinese herbal medicine, the same herb such as licorice can get different chemical constituents through different processing technologies [56]. Therefore, many researchers have focused on the pharmacological effects of raw, roasted and honey-roasted licorice. After honey-roasting, the contents of the effective chemical components varied, with lessening of the decocting quantity of liquiritin apioside and liquiritin and increase in isoliquiritin [19,25]. After roasted processing, thermal conversion did occur in the roasted licorice, indicating the conversion of glycyrrhizin to glycyrrhetinic acid [10]. Also non-polar compounds containing glycyrrhizin-degraded products, such as glycyrrhetinic acid and glycyrrhetinic acid monoglucuronide, were increased in roasted licorice in liquid chromatography analysis [12].This finding may shed some light on understanding the differences in the therapeutic values of raw, roasted and honey-roasted licorice in herbal medicine. Previous studies demonstrated that roasted licorice has been used rather than raw licorice. The roasting process of licorice results the conversion of glycyrrhizin to glycyrrhetinic acid [10,12], and thus, reduces the levels of glycyrrhizin [11]. Moreover, it has been shown that roasted licorice is more potent than un-roasted licorice in inhibiting allergic [11] and inflammatory responses [14].

Summarizing the results of this article, RGP can induce apoptosis of glioma cells and arrest the cell cycle; these two strategies are the ways to inhibit cancerous tumors (Figure 10). RGP inhibits cell proliferation and induces caspase-3-related apoptosis. In addition, RGP also induces cell cycle arrest through mediating cell cycle-related regulatory proteins, thereby strengthening the G2 checkpoint supervision function and leading to G2/M phase cell accumulation. Therefore, RGP is an anticancer drug with dual effects of inducing auto-apoptosis and inducing cell cycle arrest.

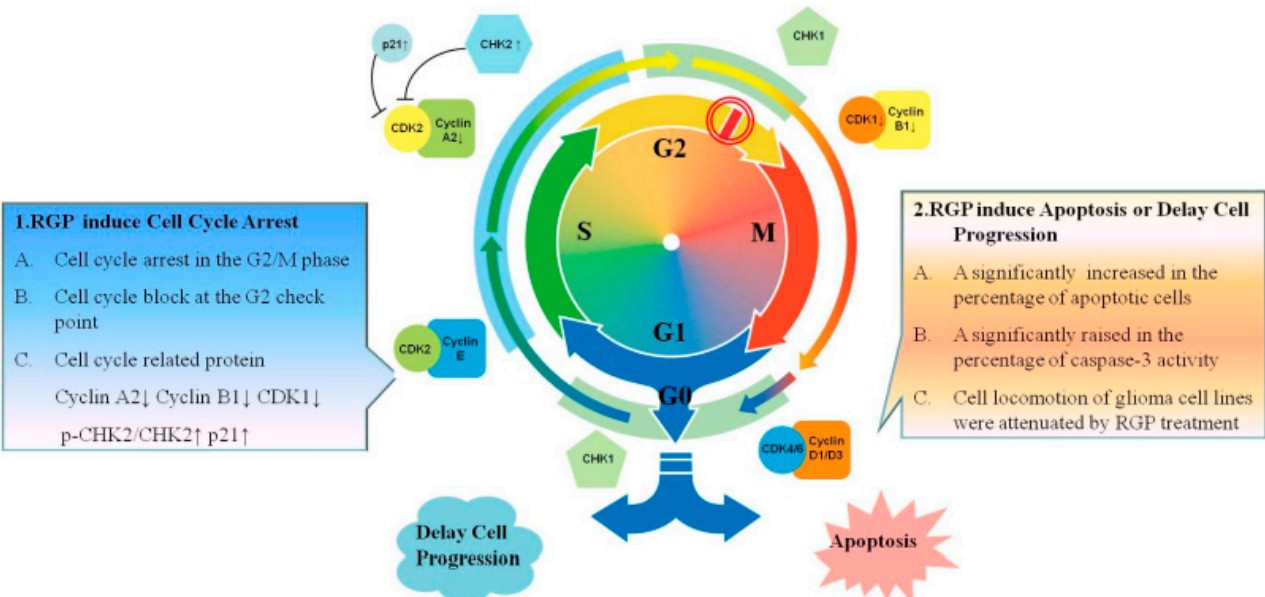

**Figure 10.** RGP induces apoptosis and G2/M cell cycle arrest in Glioblastoma Multiforme. RGP induces G2/M phase cell cycle arrest by activating the DNA checkpoint system and cell cycle regulators; and induces apoptosis in Glioblastoma Multiforme. G1 phase: Growth 1 phase. S phase: Synthesis phase. G2 phase: Growth 2 phase. M phase: mitosis. CDK1: Cyclin Dependent Kinase 1. CDK2: Cyclin Dependent Kinase 2. CHK1: Checkpoint Kinase 1. CHK2: Checkpoint Kinase 2.

## 5. Conclusions

In conclusion, RGP attenuated glioma cell viability and induced apoptosis, and inhibited cell locomotion in vitro. In addition, RGP could induce glioma cell cycle arrest and G2/M phase accumulation by mediating cell cycle-related proteins. These findings demonstrate that RGP can inhibit GBM cell proliferation, cell locomotion, and cell cycle progression, and induce apoptosis in vitro.

**Author Contributions:** Conceptualization, Y.-C.H. and T.-H.T.; methodology, Y.-C.H.; software, K.-T.L.; validation, Y.-C.H., T.-H.T. and T.-Y.L.; formal analysis, R.-D.T.; investigation, T.-Y.L.; resources, Y.-C.H.; data curation, T.-Y.L. and T.-H.W.; writing—original draft preparation, T.-Y.L. and T.-H.W.; writing—review and editing, Y.-C.H., T.-H.T. and K.-T.L. All authors have read and agreed to the published version of the manuscript.

**Funding:** This work was supported by the Kaohsiung Municipal Ta-Tung Hospital, Kaohsiung, Taiwan (KMTTH 110TA-04 and KMTTH 110-009). The funders had no role in study design, data collection and analysis, decision to publish, or preparation of the manuscript.

**Institutional Review Board Statement:** Not applicable.

**Informed Consent Statement:** Not applicable.

**Data Availability Statement:** The data used to support the findings of this study are available in article.

**Acknowledgments:** All authors thank Department of Surgery, Kaohsiung Municipal Ta-Tung Hospital, Kaohsiung, Taiwan, funding and Kuan-Ting Lee for this work.

**Conflicts of Interest:** The authors declare no conflict of interest.

## Abbreviations

| | |
|---|---|
| (Apaf-1) | Apoptotic protease activating factor 1 |
| (Bax) | BCL2 Associated X |
| (Bcl-2) | B-cell lymphoma 2 |
| (CDK2) | Cyclin-dependent kinase 2 |
| (CDK4) | Cyclin-dependent kinase 4 |
| (CHK1) | Checkpoint Kinase 1 |
| (CHK2) | Checkpoint Kinase 2 |
| (ERK) | Extracellular signal-regulated kinase |
| (FADD) | Fas-associated protein with death domain |
| (GBM) | Glioblastoma multiforme |
| (GA) | Glycyrrhizic acid |
| (HMGB1) | High Mobility Group Protein 1 |
| (IAA) | Isoangustone A |
| (ILQ) | Iso-Liquiritigenin |
| (LicA) | Licochalcone A |
| (LQ) | Liquiritin |
| (MG) | Malignant Glioma |
| (MAPK) | Mitogen-activated protein kinase |
| (NF-κB) | Nuclear factor-κB |
| (PI3K) | Phosphatidylinositol 3-kinase |
| (PARP) | Poly ADP-ribose polymerase |
| (RG) | Radix Glycyrrhizae |
| (RGP) | Radix Glycyrrhizae Preparata |
| (Rb) | Retinoblastoma protein |
| (WHO) | World Health Organization |

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
