# Peer review of "Radix Glycyrrhizae Preparata Induces Cell Cycle Arrest and Induced Caspase-Dependent Apoptosis in Glioblastoma Multiforme"

_2035-8377, doi:10.3390/neurolint14040066_

Round 1

Reviewer 1 Report

This is an original paper about the potential treatment of Glioblastoma multiforme using Radix Glycyrrhizae Preparata. This paper presents exciting results. This paper has the potential to be published. However, it has some shortcomings, and errors, and some parts are not well understood.  Hence, enhancing the quality of writing is highly recommended.

Other concerns are:

Line 81: “Thus, it is in our belief that as genetic research of the biologic nature of glioblastoma develops, new biomarkers and therapeutic targets will emerge”. This is not well understood

Line 88: RG, is one of the commonly used and old herbs in the East and West, and has been used since ancient times as 89 a useful drug in traditional medicine. Please add the reference.

Line 103: “Even more exciting is that RGP has been found to inhibit certain cancers including lung cancer”. please rewrite and add references.

Line 104: Please delete this “As far as I know”

You may enhance the quality of your introduction by explaining the molecular pathways dysregulated in GBM especially those reported by your study, and by providing more evidence about RGP and other relevant references.

Please mention each component with the doses used (For example DMSO), and please add references for each protocol used.

Concerning the discussion section, I suggest rewriting it and explaining your results, and supporting them with previous studies on glioblastomas if there are studies that have used RG that will be good. 

Author Response

Dear Editor:

Thank you for giving me the opportunity to submit a revised draft of our manuscript which titled: Radix Glycyrrhizae Preparata induces cell cycle arrest and induced caspase-dependent apoptosis in Glioblastoma Multiforme. We appreciate the time and effort that you and the reviewers have dedicated to providing your valuable feedback on our manuscript. We are grateful to the reviewers for their insightful comments on this paper. We have been able to incorporate changes to reflect most of the suggestions provided by the reviewers. We have highlighted the modification within the manuscript and point-by-point response to the reviewers’ comments and concerns in this revision.

Other concerns are:

Comments from Reviewer 1

This is an original paper about the potential treatment of Glioblastoma multiforme using Radix Glycyrrhizae Preparata. This paper presents exciting results. This paper has the potential to be published. However, it has some shortcomings, and errors, and some parts are not well understood.  Hence, enhancing the quality of writing is highly recommended.

Comment 1:

Line 81: “Thus, it is in our belief that as genetic research of the biologic nature of glioblastoma develops, new biomarkers and therapeutic targets will emerge”. This is not well understood

Response to the comment 1:

Thank you for pointing this out. We agree with this comment. I have changed "Thus, it is in our belief that as genetic research of the biologic nature of glioblastoma develops, new biomarkers and therapeutic targets will emerge" to "Therefore, we should be more active in genetic research on the biological properties of glioblastoma, and expect to develop more potential biomarkers and therapeutic targets(Page4 line8 – Page4 line9 ).

Comment 2: Line 88: RG, is one of the commonly used and old herbs in the East and West, and has been used since ancient times as a useful drug in traditional medicine. Please add the reference.

Response to the comment 2:

Agree. We have, accordingly modified the manuscripts and results to emphasize this point. We have added references and citations in the article (Page4 line10 – Page4 line10 ).

  1. Fiore C, Eisenhut M, Ragazzi E, Zanchin G, Armanini D. A history of the therapeutic use of liquorice in Europe. J Ethnopharmacol. 2005 Jul 14;99(3):317-24. doi: 10.1016/j.jep.2005.04.015. PMID: 15978760; PMCID: PMC7125727.
  2. Hayashi H, Yokoshima K, Chiba R, Fujii I, Fattokhov I, Saidov M. Field Survey of Glycyrrhiza Plants in Central Asia (5). Chemical Characterization of G. bucharica Collected in Tajikistan. Chem Pharm Bull (Tokyo). 2019;67(6):534-539. doi: 10.1248/cpb.c18-00881. PMID: 31155558.

Comment 3: Line 103: “Even more exciting is that RGP has been found to inhibit certain cancers including lung cancer”. Please rewrite and add references.

Response to the comment 3:

Thank you for pointing this out. We agree with this comment. We have rewritten "Even more exciting is that RGP has been found to inhibit certain cancers including lung cancer" to "Notably, RGP inhibits certain cancers, including lung cancer. But research on the effects of RPGs on gliomas is rather limited, and it is unclear what effects RGPs might have on GBM and how it achieves its effects." and add references and citations in the article. (Page5 line7 – Page5 line8 )

  1. Shen HS, Wen SH. Effect of early use of Chinese herbal products on mortality rate in patients with lung cancer. J Ethnopharmacol. 2018 Jan 30;211:1-8. doi: 10.1016/j.jep.2017.09.025. Epub 2017 Sep 20. PMID: 28942131.

Comment 4: Line 104: Please delete this “As far as I know”

Response to the comment 4:

Thank you for this suggestion. It would have been interesting to explore this aspect. We have removed "As far as I know" from the article. We have rewritten "As far as I know, research on the effects of RPGs on gliomas is rather limited, and it is unclear what effects RGPs might have on GBM and how it achieves its effects." But research on the effects of RPGs on gliomas is rather limited, and it is unclear what effects RGPs might have on GBM and how it achieves its effects.” (Page5 line8 – Page5 line9 ).

Comment 5: You may enhance the quality of your introduction by explaining the molecular pathways dysregulated in GBM especially those reported by your study, and by providing more evidence about RGP and other relevant references.

Response to the comment 5:

Thank you for this suggestion. We have modified and provided more evidence about RGP and other relevant references. We have provided more evidence on RGP and other relevant references in the introduction to improve the quality of your presentation. And explain this introduction as the mechanism of RGP's anti-cancer, so as to understand the purpose of the author's use of the experiment (Page4 line10 – Page5 line6 ).

Comment 6: Please mention each component with the doses used (For example DMSO), and please add references for each protocol used.

Response to the comment 6:

Agree. We had been revised the title. Thank you for this suggestion. We have added references for each of our protocols to the Materials and Methods section of the article.

  1. Cell viability: Mosmann T (1983) Rapid colorimetric assay for cellular growth and survival: application to proliferation and cytotoxicity assays. Journal of immunological methods 65(1-2):55-63 doi:10.1016/0022-1759(83)90303-4. (Page6 line4 – Page6 line5 ).
  2. Cell cycle analysis: Riccardi, C. and I. Nicoletti, Analysis of apoptosis by propidium iodide staining and flow cytometry. Nat Protoc, 2006. 1(3): p. 1458-61. (Page6 line27 – Page6 line28 ).
  3. Apoptosis assessment: (1)Casciola-Rosen, L., et al., Surface blebs on apoptotic cells are sites of enhanced procoagulant activity: implications for coagulation events and antigenic spread in systemic lupus erythematosus. Proc Natl Acad Sci U S A, 1996. 93(4): p. 1624-9.(2) van Engeland, M., et al., A novel assay to measure loss of plasma membrane asymmetry during apoptosis of adherent cells in culture. Cytometry, 1996. 24(2): p. 131-9. (Page7 line21 – Page7 line22 ).
  4. Caspase 3 activity assay: (1)Chen P-C, et al. Nrf2-mediated neuroprotection in the MPTP mouse model of Parkinson’s disease: critical role for the astrocyte. Proc Natl Acad Sci. 2009;106(8):2933–8.(2) Wu Y, et al. ROCK inhibitor Y27632 promotes proliferation and diminishes apoptosis of marmoset induced pluripotent stem cells by suppressing expression and activity of caspase 3. Theriogenology. 2016;85(2):302–14. (Page8 line3 – Page8 line4 ).
  5. WesternBlot: Zheng W-H, Quirion R. Glutamate acting on N-methyl-D-aspartate receptors attenuates insulin-like growth factor-1 receptor tyrosine phosphorylation and its survival signaling properties in rat hippocampal neurons. J Biol Chem. 2009;284(2):855–61. (Page9 line13).
  6. Mitotic index analysis: J Biol Chem 274, 25543 (1999); 2. Chromosome Res 14, 393 (2006); 3. Nature 438, 1176 (2005). (Page10 line3 - Page 10line5 ).

Comment 7: Concerning the discussion section, I suggest rewriting it and explaining studies that have used RG that will be good.

Response to the comment 7: Thanks to the reviewer for pointing out this question. Agree. We have modified the manuscript to emphasize this point. We have rewritten it and explained that research using RG would be good. Also we try to minimize grammatical structural errors and typos as possible and typical English was used in this article. (Page13 line19 – Page16 line29 ).

Sincerely,

Tai-Hsin Tsai, MD, PhD.

Department of Neurosurgery

Kaohsiung Medical University Hospital

No. 100, Tzyou 1st Road, Sham-min District

Kaohsiung City, Taiwan

E-mail:[email protected]

Reviewer 2 Report

The present manuscript is interesting and the role played by RGP and its main components will certainly have relevance in cancer therapy. However, the authors of this manuscript need to demonstrate why they used high concentrations of RGP (0.5-2 mg / ml) in their experiments. By now, the possible interaction of molecules or plant extracts with the cell culture medium, with the production of pro-oxidant molecules, which may be responsible for the observed biological effects, is well described in the scientific literature. In this regard, I ask the authors to perform a FOX assay for the identification in the cell culture medium of hydrogen peroxide, or to incubate RGP at various concentrations with the enzyme catalase (at least 100 U), to verify or not the involvement of these reactive molecules in your results. Also, I don't understand why you have included paragraphs dedicated to the main active ingredients of Radix Glycyrrhizae. Have you isolated the active components (Glycyrrhizic acid and Liquiritin)? If so, why did you not present any experimental data in this paper? Otherwise, if you know the order of magnitude of their concentrations in your extract, you need to make a dose-effect curve using commercial pure molecules. What is the role of ROS in the biological effects of RGP ? Have you measured intracellular peroxide levels with DCFH-DA? Finally, have you made a measurement of the concentration of polyphenols (Folin assay), to see if the biological activity manifested is related to the total polyphenols?

Author Response

Journal Neurology International (ISSN 2035-8377)

Manuscript ID: neurolint-1916138

Type: Article

Title: Radix Glycyrrhizae Preparata induces cell cycle arrest and induced caspase-dependent apoptosis in Glioblastoma Multiforme

Dear Editor:

Thank you for giving me the opportunity to submit a revised draft of our manuscript which titled: Radix Glycyrrhizae Preparata induces cell cycle arrest and induced caspase-dependent apoptosis in Glioblastoma Multiforme. We appreciate the time and effort that you and the reviewers have dedicated to providing your valuable feedback on our manuscript. We are grateful to the reviewers for their insightful comments on this paper. We have been able to incorporate changes to reflect most of the suggestions provided by the reviewers. We have highlighted the modification within the manuscript and point-by-point response to the reviewers’ comments and concerns in this revision.

Other concerns are:

Comments from Reviewer 2:

The present manuscript is interesting and the role played by RGP and its main components will certainly have relevance in cancer therapy.

Comment 1: The authors of this manuscript need to demonstrate why they used high concentrations of RGP (0.5-2 mg / ml) in their experiments.

Response to the comment 1:

Thanks for pointing this out. In this experiment, we used RGP to treat GBM cells, we calculated ID50 during pretest. The inhibitory effects of RGP were similar between the GBM8401 and U87MG cells. The growth inhibitory dose of 50% (ID50) of RGP was 3.88 mg/ml in GBM8401 cells and 5.26 mg/ml in U87MG cells, respectively. And using the ID50 of these two cell lines to set the concentration in our experiments, we used high concentrations of RGP (0.5-2 mg/ml) in their experiments (Page11 line3 – Page11 line5 ).

Comment 2: The possible interaction of molecules or plant extracts with the cell culture medium, with the production of pro-oxidant molecules, which may be responsible for the observed biological effects, is well described in the scientific literature.

Response to the comment 2:

Thank you for pointing this out. We agree with this comment. It would have been interesting to explore this aspect. We examined RGP’s effects on cell growth and cell cycle regulation and evaluated the expression levels of downstream molecules, resulting in the inability of cells to carry out cellular processes, eventually leading to cell cycle arrest or inducing apoptosis. Thank you for pointing this out. We agree with this comment. It would have been interesting to explore this aspect. We examined RGP's effects on cell growth and cell cycle regulation and evaluated the expression levels of downstream molecules, resulting in the inability of cells to carry out cellular processes, eventually leading to cell cycle arrest or inducing apoptosis. This study is not to explore the correlation between the antioxidant effect of RGP and the anticancer effect, and I would like to thank Reviewer for the opinion, the correlation between antioxidant effect and anticancer effect. This opinion will be an important direction for our research team's follow-up research (Page11 line7 – Page17 line11 ).

Comment 3: In this regard, I ask the authors to perform a FOX assay for the identification in the cell culture medium of hydrogen peroxide, or to incubate RGP at various concentrations with the enzyme catalase (at least 100 U), to verify or not the involvement of these reactive molecules in your results.

Response to the comment 3:

Thank you for pointing this out. We agree with this comment. The present study focuses on RGP and its chemopreventive activities against glioma cells. We examined RGP’s effects on cell growth and cell cycle regulation and evaluated the expression levels of downstream molecules, resulting in the inability of cells to carry out cellular processes, eventually leading to cell cycle arrest or inducing apoptosis. This study is not to explore the correlation between the antioxidant effect of RGP and the anticancer effect. Therefore, we didn't schedule the Fox assay. I would like to thank Reviewer for the opinion, the correlation between antioxidant effect and anticancer effect. This opinion will be an important direction for our research team's follow-up research (Page11 line7 – Page17 line11 )..

Comment 4: Also, I don't understand why you have included paragraphs dedicated to the main active ingredients of Radix Glycyrrhizae. Have you isolated the active components (Glycyrrhizic acid and Liquiritin)? If so, why did you not present any experimental data in this paper? Otherwise, if you know the order of magnitude of their concentrations in your extract, you need to make a dose-effect curve using commercial pure molecules.

Response to the comment 4:

Thank you for this suggestion. We have a paragraph dedicated to the main active ingredients of licorice in the Discussion section, which we have revised in the article. This paragraph is mainly written because RGP is not a purified substance, so it will achieve the effect of medicine because of its active ingredients, so the active ingredient of licorice is specially introduced.We also think that it is really inappropriate to introduce licorice actives in the discussion section, so it has been revised in the article.We did not isolate or purify the active ingredients (glycyrrhizic acid and liquiritin) of licorice in this experiment and therefore do not provide experimental data in the article.

Comment 5: What is the role of ROS in the biological effects of RGP ? Have you measured intracellular peroxide levels with DCFH-DA?

Response to the comment 5:

Thank you for this suggestion. Thank you for pointing this out. We agree with this comment. The present study focuses on RGP and its chemopreventive activities against glioma cells. We examined RGP’s effects on cell growth and cell cycle regulation and evaluated the expression levels of downstream molecules, resulting in the inability of cells to carry out cellular processes, eventually leading to cell cycle arrest or inducing apoptosis. Our study is not to investigate the anti-oxidant and anti-cancer effects of this drug, so we did not explore the role of ROS in the biological effects of RGP nor did we measure intracellular peroxide levels with DCFH-DA. I would like to thank Reviewer for the opinion, the correlation between antioxidant effect and anticancer effect. This opinion will be an important direction for our research team's follow-up research (Page11 line7 – Page17 line11 ).

Comment 6: Finally, have you made a measurement of the concentration of polyphenols (Folin assay), to see if the biological activity manifested is related to the total polyphenols?

Response to the comment 6:

Thank you for this suggestion. Thank you for pointing this out. We agree with this comment. The present study focuses on RGP and its chemopreventive activities against glioma cells. We examined RGP’s effects on cell growth and cell cycle regulation and evaluated the expression levels of downstream molecules, resulting in the inability of cells to carry out cellular processes, eventually leading to cell cycle arrest or inducing apoptosis. Our study is not to investigate the anti-oxidative and anti-cancer effects of this drug, so we did not explore the measurement of the concentration of polyphenols (Folin assay) and neither, to see if the biological activity manifested is related to the total polyphenols. I would like to thank Reviewer for the opinion, the correlation between antioxidant effect and anticancer effect. This opinion will be an important direction for our research team's follow-up research (Page11 line7 – Page17 line11 )..

Additional clarifications

In addition to the above comments, all spelling and grammatical errors pointed out by the reviewers have been corrected.

Sincerely,

Tai-Hsin Tsai, MD, PhD.

Department of Neurosurgery

Kaohsiung Medical University Hospital

No. 100, Tzyou 1st Road, Sham-min District

Kaohsiung City, Taiwan

E-mail:[email protected]

Round 2

Reviewer 1 Report

This is a relevant paper that highlights the potential role of Radix Glycyrrhizae Preparata (RGP) against cancerous tumors. Based on these results, RGP can inhibit Glioblastoma multiforme cell proliferation, and cell locomotion, and induce apoptosis in vitro. This paper is generally well-written. However, I suggest English proofreading. 

Reviewer 2 Report

I thank the authors for answering my questions